

# Methanogenesis by $CO_2$ reduction dominates lake sediments with different organic matter compositions

Guangyi Su[1*], Julie Tolu[2], Clemens Glombitza[3], Jakob Zopfi[4], Moritz F. Lehmann[4], Mark A. Lever[3], Carsten J. Schubert[1, 3]

[1] Swiss Federal Institute of Aquatic Science and Technology (EAWAG), Department of Surface Waters – Research and Management, 6047 Kastanienbaum, Switzerland

[2] Swiss Federal Institute of Aquatic Science and Technology (EAWAG), Department of Water resources and Drinking water, 8600 Duebendorf, Switzerland

[3] Institute of Biogeochemistry and Pollutant Dynamics, ETH Zurich, 8092 Zurich, Switzerland

[4] Department of Environmental Sciences, University of Basel, 4056 Basel, Switzerland

[*] corresponding author: Guangyi Su, guangyi.su@unibas.ch

## Abstract

Microbial methane production is a respiration reaction involved in the terminal step of anaerobic degradation of organic matter. Due to the dependency of methanogenic substrate production on fermentation reactions that produce different end productions, different sources and compositions of organic carbon (OC) may impact the methanogenic potential in lake sediments. Here, we investigate the sources and compositions of OC in sediments of Lake Geneva and how both are potentially linked to methane production. Differences in dominant long-chain fatty acid abundances and carbon isotopic compositions suggest the predominance of diagenetically altered phytoplankton-derived OC at a profundal site and temporally highly variable sources of both aquatic and terrestrial OC in a deltaic location. Despite these differences, radiotracer-based methanogenesis rate measurements and stable isotopic signatures of methane indicate significant methane production that is dominated by $CO_2$ reduction (>95% of total methanogenesis) in both locations. Matching this interpretation, members of well-known $CO_2$-reducing *Methanoregula* sp. dominate both sites. No clear effect of OC source on methane production rates was evident. Our data demonstrate that OC of diverse sources and diagenetic states support microbial methane production, but do not indicate a clear impact of the OC source on the dominant methanogenic pathway or the community structure of methanogenic microorganisms in lacustrine sediments.



Keywords: Methane production rate; Methanogenesis pathway; Sediment organic carbon; Lipid biomarkers; Methanogen community

40

## 1. Introduction

Lakes represent an important source of methane ($CH_4$) to the atmosphere (Bastviken et al., 2011), which is a potent greenhouse gas with a global warming potential more than 27 times that of carbon dioxide on a 100-year basis (GWP-100) (Masson-Delmotte et al., 2021). A large fraction of $CH_4$ produced in lakes is produced during the anaerobic decomposition of organic carbon (OC) in sediments, from where it escapes by ebullition or diffusion into the bottom waters. Methane formation is the final step in the degradation of organic matter, and is catalyzed by anaerobic methanogenic archaea, capable of using several substrates including $H_2/CO_2$, acetate, and methylated compounds (Lyu et al., 2018). It is still poorly understood how methane production in lake sediments is regulated, and to what extent the methanogenic potential is related to OC quality. In this regard, microbial organic matter degradation reactions (e.g., hydrolysis, fermentation, and anaerobic oxidation) play a critical role by sequentially breaking down organic macromolecules, such as proteins, carbohydrates, and lipids, to acetate and $H_2/CO_2$, which are then used as substrates by methanogenic archaea (Demirel and Scherer, 2008).

Freshwater lakes cover only a small portion of the Earth's surface (< 3%) (Downing et al., 2006), compared to oceans (71%), yet, the annually accumulated OC in lakes represents nearly half of what is stored in the oceans (Mendonça et al., 2017). In many lakes, most of organic matter in sediments is derived from autochthonous aquatic organisms like phytoplankton and aquatic macrophytes (Dean and Gorham, 1998). On the other hand, allochthonous organic carbon such as detritus of terrestrial vegetation can account for a significant fraction of the lacustrine sedimentary organic matter (Larsen et al., 2011), for example within river deltas (Randlett et al., 2015). In small or oligotrophic lakes and/or high-latitude/altitude lakes, OC sedimentation may in fact be dominated by land-derived organic matter, and it has previously been observed that high carbon burial efficiency in sediments is linked to a high proportion of allochthonous OC (Sobek et al., 2009). In general, OC from aquatic biomass such as phytoplankton mainly comprises relatively labile compounds (Parsons et al., 1961). In





contrast, allochthonous OC (e.g., terrestrial plants) contains more complex structural
and biochemically recalcitrant compounds, such as cellulose and lignin (Opsahl and
Benner, 1995), which are more effectively preserved over time (Han et al., 2020, 2022).
The different biochemical compositions and characteristics of OC in lake sediments,
and the associated differential susceptibility to hydrolytic attack and microbial
breakdown into smaller carbon compounds (Lehmann et al., 2002a, 2020), might
substantially affect $CH_4$ production and emission, as has been suggested in previous
studies (Grasset et al., 2018; West et al., 2012). However, laboratory experiments
performed at short-time scales (i.e., within weeks or months), involving the spiking of
sediments with fresh algal and/or plant organic materials, do not accurately reflect
effects of enhanced OC contribution to natural sediments on $CH_4$ production because
the compositions and lability of original organic materials may vary greatly after
sedimentation due to oxidative destruction and microbial alteration during sinking in
the water column (Kawamura et al., 1987). Moreover, allochthonous OC found in lake
sediments is often the less-reactive (i.e. less bio-degradable) remnant of land-derived
debris, where the more reactive fractions have already been removed on land or during
fluvial transport (Raymond and Bauer, 2001). Lastly, while it is known that organic
matter bioavailability decreases over time, as labile components are selectively
remineralized (Grasset et al., 2018), the effects of organic matter quality and OC
degradation state, and their interactions, on $CH_4$ production remain uncertain.

Lipid biomarkers and their stable carbon isotopic composition can be used to infer

sources and diagenetic state of lacustrine organic matter (Dai et al., 2005; Meyers and
Ishiwatari, 1993). Different chain lengths of fatty acids and n-alkanes can be used for
the distinction between aquatic phytoplankton and terrestrial plants (Cranwell, 1976;
Eglinton and Hamilton, 1967). In addition, primary producers growing in different
habitats (e.g., terrestrial plants and freshwater phytoplankton) have distinct isotopic
compositions, due to the differences in C sources and biochemical pathways, through
which inorganic carbon is assimilated and incorporated into biomass (Cloern et al.,
2002). For example, terrestrial plants incorporate carbon dioxide from the atmosphere
using the C3 Calvin-Benson pathway and consequently have an average bulk isotopic
value of ca -28‰, while isotopic values of various aquatic plants are often significantly
less negative (e.g., an average of -20‰ for benthic diatoms) (Cloern et al., 2002;
O'Leary, 1981). Moreover, stable carbon isotope measurements can be used to trace
the carbon flow from organic matter degradation to $CH_4$ in lake sediments. The isotopic





signatures of methane and methane precursors (e.g., dissolved inorganic carbon, bulk
organic carbon) can be used to assess the relative contribution of the major pathways
(i.e., hydrogenotrophic and acetoclastic methanogenesis) to total $CH_4$ production
(Conrad, 2005; Whiticar, 1999). To date it remains unclear, which organic compounds
represent the main precursors of molecules that are ultimately converted to methane in
older sediments, in which more labile OC fractions, such as OC from microalgal cells,
have largely been remineralized already. Previous research in Lake Geneva revealed
higher benthic $CH_4$ fluxes, but lower total mineralization rates of organic matter in
deltaic sediments, compared to profundal sites with reduced riverine impact (Randlett
et al., 2015). The observed differences imply that despite less efficient total
remineralization that leads to elevated OC burial rates in deltaic sediments, high $CH_4$
production is sustained due to the high input of allochthonous OC (Sollberger et al.,

2014).

Here we investigated relationships between methanogenesis rates and pathways,

and the sources and degradation state of sedimentary organic matter in profundal and
deltaic sediments of Lake Geneva. We combined methanogenic rate measurements by
radiotracer incubations with $^{14}C$-labeled bicarbonate and acetate with compositional
(lipids, pyrolysis-GC/MS) and stable carbon isotopic analyses of organic carbon and
$CH_4$, and sedimentation rate measurements (based on Pb-210, Cs-137). Additional
quantitative analyses of a methanogenic marker gene (*mcr*A) and analyses of
methanogenic community structure (16S rRNA gene sequences) provided insights into
the abundances and identities of in situ methanogenic populations. Based on this multi-
disciplinary data set, we identified potential relationships between sediment OC sources
and degradation status, rates, pathways, and the organisms involved in the microbial
production of methane.

**2. Material and methods**
**2.1. Study sites**

Lake Geneva is the largest western European lake. The Rhone River, the main

tributary to the lake in the northeastern part, has a catchment area of 5220 $km^2$ and
accounts for about 68% of the total water discharge and large amounts of suspended
sediment and fine-particle loading to the lake (Burrus et al., 1989). The Rhone River
inflow brings in large amounts of terrestrial OC, which is mostly deposited near the



river mouth as an important contribution to deltaic sediments. On the other hand,
sedimentation in the deeper part of the lake is usually dominated by phytoplankton-
derived OC (Gallina et al., 2017).

**2.2. Sample collection and processing**
Sediment cores were collected using a larger gravity corer (14 cm inner diameter)
at a profundal site (46°25′54″N, 6°47′33″E, water depth: 240 m) and a smaller gravity
corer (6.5 cm inner diameter) at a deltaic site (46°24′58″N, 6°51′34″E, water depth: 128
m) in August and December 2019, respectively. Profundal samples for different
analyses were obtained from a single sediment core at 2-cm vertical resolution. Samples
for the analysis of dissolved methane concentrations were collected on site with cut-off
syringes through pre-drilled holes in the core liner covered with adhesive tape.
Sediment samples of 2 cm$^3$ were fixed with 5 mL 10% NaOH in 120 mL serum bottles,
which were capped immediately with thick butyl rubber stoppers and an aluminum
crimp cap. Additionally, two replicate samples were taken from the same depth for
methanogenesis rate measurements (described below). Sediment porewater was
extracted with Rhizon samplers (Rhizosphere Research Products, Wageningen,
Netherlands) connected to 20-mL syringes through predrilled small holes, with 2 cm
distance between them. Porewater samples for DIC concentration measurements and
stable carbon isotope analyses were stored in 4-mL glass vials without headspace at 4
°C. Separate porewater sample aliquots for the quantification of acetate and other
volatile fatty acids were stored in combusted (450 °C for 5 h) glass vials at -20 °C until
further analysis. The sediment core was then extruded in the lab and sectioned into 2
cm segments. Samples were taken from each segment and stored frozen (-20 °C) until
further analysis for bulk parameters and lipid biomarkers. Using the gravity corer
equipped with the smaller core liner, four sediment cores were obtained from the deltaic
site. One core was used for methane concentration measurements, a second one for rate
measurements (both at 2-cm resolution, sample collection as described above). The
third core was used for porewater extraction in the lab using Rhizon samplers, and the
last one was split open for the determination of porosity, analysis of bulk parameters
and lipid biomarkers. An additional small-diameter sediment core was taken at the
profundal site during the second sampling campaign for porosity analysis.

**2.3. Rate measurements of methanogenesis using radiolabeled substrates**



Methanogenesis rates (MGR) were determined using a radioisotope-based
approach. We determined the modes of methane production and activity rates in
incubation experiments with radio-labeled acetate and bicarbonate. At each depth,
samples (2.5 cm$^3$) were collected through pre-drilled holes using 3 mL cut-off plastic
syringes, which were closed with rubber stoppers and stored at 4 °C. Upon arrival of
the samples in the laboratory, ten microliters of anoxic $^{14}$C-labeled bicarbonate (~16
kBq, Perkin-Elmer) or 2-$^{14}$C-labeled (i.e., 14C label in the carboxyl group) acetate
solution (~18 kBq, Perkin-Elmer) were injected for the MGR measurements to
determine hydrogenotrophic (MGR$_{DIC}$) and acetoclastic (MGR$_{Ac}$) methanogenesis
rates, respectively. Immediately after the tracer injection through the stoppers, all
samples including killed controls (i.e., samples that were transferred to 10 mL 5%
NaOH solution immediately after tracer injection) were incubated under an N$_2$
atmosphere at in situ temperature (4 °C) in the dark for 48 h.
To stop microbial activity in the incubations with $^{14}$C-labeled substrates, samples
were transferred into 120-mL serum bottles containing 10 mL aqueous NaOH (5%
wt:wt), immediately crimp-sealed with butyl rubber stoppers and vigorously shaken.
The $^{14}$C activity in the different carbon pools was determined as previously described
(Su et al., 2019). In brief, the headspace of a fixed sample is purged with air (30 mL
min$^{-1}$ for 30 min) through a heated (850 °C) quartz tube filled with copper oxide, where
the $^{14}$CH$_4$ (product of methanogenesis during incubation) is combusted to $^{14}$CO$_2$. The
$^{14}$CO$_2$ is then captured in two sequential traps of scintillation vials (20 mL) containing
8 mL of 1:7 phenylethylamine and methoxyethanol. The cumulative radioactivity of
both traps is then determined by liquid scintillation counting (2200CA Tri-Carb Liquid
Scintillation Analyzer) after adding 8 mL of a scintillation cocktail (Ultima Gold,
PerkinElmer) to each vial, and thorough mixing using a vortex-mixer. For incubations
with bicarbonate, the residual $^{14}$C-bicarbonate was measured as $^{14}$CO$_2$, released from
the alkaline liquid phase after adding 2.5 mL of 32% HCl. The remaining radioactivity
(possibly explained by inorganic carbon assimilation into biomass) was determined in
a 1-mL aliquot of the acidified mixture (amended with 4 mL Ultima Gold) by liquid
scintillation counting. Incubation bottles with $^{14}$C-acetate were also acidified with 2.5
mL of 32% HCl after extraction of $^{14}$CH$_4$, and $^{14}$CO$_2$ from microbial acetate oxidation
was subsequently purged and trapped as described above. To determine the residual
$^{14}$C-acetate in the incubation vial, 1 mL of the acidified mixture was mixed with 4 mL
Ultima Gold for scintillation counting (Beulig et al., 2018). The control samples were



processed in the same way as the incubated samples after the termination of incubation.
The methanogenesis rates with DIC (nmol cm$^{-3}$ d$^{-1}$) and acetate (nmol cm$^{-3}$ d$^{-1}$) as
substrates were calculated using Eq. 1 and Eq. 2, respectively, modified from a previous
study (Beulig et al., 2018).
$$MGR_{DIC} = 1.08 \times \varphi \times [DIC] \times \frac{A_{CH_4}}{A_{CH_4} + A_{DIC} + A_R} \times t^{-1} \qquad (1)$$

$$MGR_{Ac} = 1.08 \times \varphi \times [Ac] \times \frac{A_{CH_4}}{A_{CH_4} + A_{DIC} + A_{Ac}} \times t^{-1} \qquad (2)$$

[$DIC$] and [$Ac$] are concentrations of DIC and acetate in the sediment porewater,
respectively. $A_{CH_4}$ and $A_{DIC}$ represent the activities of produced CH$_4$ and DIC (i.e.,
residual DIC in bicarbonate-amended incubations, product DIC in acetate-amended
incubations) at the end of the incubation (in CPM). $A_R$ and $A_{Ac}$ represent the remaining
radioactivity of DIC-incubated samples (i.e., biomass and metabolic intermediates) and
the residual $^{14}$C-acetate radioactivity (possibly also including a small fraction of $^{14}$C in
metabolic intermediates and incorporated into biomass), respectively. $\varphi$ is the porosity
of the sediment samples. The factor 1.08 accounts for the isotopic fractionation of $^{14}$C
(Hansen et al., 2001). $t$ represents the incubation time in days. Measured $^{14}$CH$_4$
activities in all incubation samples were blank-corrected by subtracting the $^{14}$CH$_4$
activity measured in the killed control (typically close to background radioactivity)
incubated with the same amounts of $^{14}$C-labeled substrates. The activities in the $^{14}$CH$_4$
pool were considered zero if the blank-corrected value was negative.

**2.4. Pore water methane and nutrient analyses**
Methane concentrations were measured in the headspace of NaOH-preserved
samples using a gas chromatograph (GC, Agilent 6890N) with a flame ionization
detector, and helium as a carrier gas. CH$_4$ concentrations in the wet sediments were
then calculated based on the headspace-to-sample volume ratio. The C isotopic
composition ($^{13}$C/$^{12}$C) of methane was determined in the same samples using a pre-
concentration unit (TraceGas, Micromass, UK) connected to an isotope ratio mass
spectrometer (IRMS; GV Instruments, Isoprime). Stable C-isotope values are reported
in the conventional δ notation (in ‰) relative to the Vienna Pee Dee Belemnite standard
(V-PDB), with a reproducibility of 0.5‰ based on replicate measurements of methane
standards. A carbon analyzer (TOC-L, Shimadzu, Kyoto, Japan) was used to quantify
dissolved inorganic carbon (DIC) concentrations in sediment porewaters. Samples were



manually injected into the DIC reaction vessel and measured with a non-dispersive
infrared detector (NDIR) after acidification and volatilization to $CO_2$. To determine the
carbon isotopic composition of DIC, a porewater aliquot of 1-2 mL was introduced into
a He-purged exetainer (Labco Ltd) and acidified with ~200 µL 85% $H_3PO_4$. After 2 h
equilibration at 37 °C, the $CO_2$ released from the aqueous phase was subsequently
analyzed using a preparation system (MultiFlow, Isoprime) coupled to an IRMS
(Micromass, Isoprime). The standard deviation for replicate measurements of samples
and standards was < 0.2‰. Porewater samples for acetate and other volatile fatty acids
were analyzed with a two-dimensional ion chromatography (2D IC), as described
previously (Glombitza et al., 2014), with modifications reported in (Schaedler et al.,
2018). Samples were filtered through pre-washed (10 mL Ultrapure Type I water)
disposable syringe filters (Acrodisc® IC grade, 0.2 µm pore size, 13 mm diameter).
The first 0.5 mL sample after filtration was discarded before collecting the samples for
2D IC analysis. We used a dual Dionex ICS6000 instrument (Thermo Scientific)
equipped with a Dionex AS24 column (2 mm diameter) for the first dimension, and a
Dionex AS11HC column (2 mm diameter) for the second dimension. Quantification
was done from the conductivity detector signal with a series of 5 external standards
between 0.5 and 100 µM.
**2.5. Bulk sediment analyses**
The total organic carbon (TOC) contents of sediment samples were determined by
the difference between total carbon and total inorganic carbon.  Samples for total carbon
were measured with an Elementar Vario Pyro Cube CN elemental analyser (Elementar,
Germany), and samples for total inorganic carbon were analysed by coulometry using
a UIC CM5015 coulometer (Joliet, IL, USA). Sediment cores were dated by gamma
spectrometry using $^{137}$Cs on freeze-dried and ground sediment for the determination of
sedimentation rates, as described previously (Randlett et al., 2015). The $\delta^{13}$C-TOC was
determined on decalcified samples (Schubert and Nielsen, 2000). Briefly, freeze-dried
and homogenized samples were treated with 5 mL 10% HCl in 15 mL Falcon tubes
overnight. After centrifugation, the supernatant was discarded, and the solid phase was
washed/centrifuged three times with 5 mL MilliQ water. Samples were dried at 50 °C
for 48 h prior to analysis. The $\delta^{13}$C-TOC was then assessed by elemental analysis-
isotope ratio mass spectrometry (EA, Pyro Cube, Elementar and IRMS, Isoprime, UK).



The reproducibility based on replicate measurements of standards and samples was
better than 0.2‰.

**2.6. Pyrolysis gas chromatography-mass spectrometry**
The sedimentary organic matter composition was characterized at the molecular
level using a pyrolyzer equipped with an autosampler (EGA/PY-3030D and AS-1020E,
FrontierLabs, Japan) connected to a gas chromatograph (Trace 1310, Thermo
Scientific) and a mass spectrometer (ISQ 7000, Thermo Scientific), following the
optimized method (Tolu et al., 2015). Depending on the sample, an aliquot of 2-3 mg
of dry sediment was pyrolyzed at 450°C. A data-processing pipeline, including
chromatogram smoothing, alignment background correction and multivariate curve
resolution by alternate regression was used to automatically detect and integrate the
peaks and extract their mass spectra under "R" computational environment (R Core
Team 2014). To optimize the number of detected peaks, data processing was performed
independently for the sediments from the deltaic site (DS) and the profundal site (PS).
Individual peaks were identified using "NIST MS Search 2" software which includes
the library "NIST/EPA/NIH 2011", complemented by spectra from published studies.
The relative abundances of these identified pyrolytic organic compounds were
calculated for each sample by normalization to the sum of their peak areas set at 100%.

**2.7. Lipid extraction, separation, and quantification**
Based on methanogenesis rate and geochemical profiles, sediment samples from
five selected depths of both locations were lyophilized and homogenized for subsequent
lipid extraction. Prior to extraction, an internal standard mix (5$\alpha$-androstane, 3-
eicosanone, n-C19:0 fatty acid and n-C19 alkanol) was added to each sample for the
quantification of single biomarkers. Lipids were extracted in 20 mL of 7:3
Dichloromethane/Methanol (DCM/MeOH) in a Microwave Reaction System (SolvPro,
Anton Paar, Graz, Austria), as described previously (Ladd et al., 2018). Total lipid
extracts (TLEs) were obtained by successively rinsing the samples with DCM after
centrifugation, and then concentrated using a Multivapor P-6 (Büchi Labortechnik AG,
Switzerland). TLEs were further evaporated to dryness, and saponified in 3 mL
methanolic KOH-solution (~1 N) at 80 °C for 3 h. Neutral compounds were extracted
by liquid-liquid extraction using hexane, fatty acids (FAs) were then extracted from the
remaining aqueous phase with hexane after acidification (pH < 2). The fatty acid



fraction was treated using 1 mL of BF$_3$ in methanol (14% v/v, Sigma Aldrich) at 80 °C
for 2 h, and converted to fatty acid methyl esters (FAMEs). Neutral compounds were
further separated into four different fractions using 500 mg/6mL pre-packed Si gel
columns (filling quantity/volume, Biotage, Uppsala, Sweden). Briefly, the neutral
fraction was dissolved in 4 mL hexane and transferred onto the column, followed by 4
ml hexane/DCM (2:1 v/v), then 4 mL DCM/MeOH (19:1 v/v) and finally 4 mL MeOH,
with the elution of hydrocarbon, ketone, alcohol and remaining polar compounds,
respectively. The alcohol fraction was acetylated in 25 µL acetic anhydride and 200 µL
pyridine at 70 °C for 30 min.

All fractions were quantified using a gas chromatograph equipped with a flame

ionization detector (GC-FID) (GC-2010 Plus, Shimadzu, Japan). Samples were injected
onto an InertCap 5MS/NP column (0.25 mm × 30 m × 0.25 µm, GL Sciences, Japan)
using an AOC-20i autosampler (Shimadzu) through a split/splitless injector operated in
splitless mode at 280 °C. The column was heated from 70 °C to 130 °C at 20 °C min$^{-1}$,
then to 320 °C at 4 °C min$^{-1}$, and held at 320 °C for 20 min. FAMEs were identified by
comparing their retention times to those of laboratory standards (i.e., a fatty acid methyl
ester mix and bacterial acid methyl ester from Supelco, reference no. 47885-U and
47080-U, respectively), and were quantified by normalization to the internal n-C19:0
fatty acid standard. Identification of hydrocarbons was performed by comparing their
retention times to those of an external standard containing C14 to C40 n-alkanes
(Sigma-Aldrich). The alcohols were characterized using a gas chromatography-mass
spectrometer (GC-MS, QP2020, Shimadzu, Japan) under identical chromatographic
conditions. Acquired mass spectra were identified through comparison with published
data.

### 2.8. Compound-specific stable carbon isotope analysis

The stable carbon isotope composition of FAMEs, n-alkanes and alcohols was

determined by gas chromatography-isotope ratio mass spectrometry (GC-IRMS), using
a Delta V Advantage IRMS (Thermo Scientific) with a ConFlow IV (Thermo
Scientific). Samples were injected with a TriPlus RSH autosampler to a PTV inlet
operated in splitless mode at 280 °C on a GC-1310 gas chromatograph (Thermo
Scientific, Bremen, Germany). The GC was equipped with a 30 m DB-5MS fused silica
capillary column (0.25 mm i.d., 0.25 µm film thickness). The GC oven was heated from
80 °C to 215 °C at 15 °C min$^{-1}$, then to 320 °C at 5 °C min$^{-1}$, and held at 320 °C for 10



min. Column effluent was combusted at 1020 °C. Compound-specific $\delta^{13}C$ values were
reported relative to the V-PDB scale, and calibrated externally using known $\delta^{13}C$ values
of an alkane mixture (n-C$_{17, 19, 21, 23, 25, 28 \text{ and } 34}$, Arndt Schimmelmann, Indiana
University, USA), which were run at the beginning and the end of each sequence, as
well as after every 6th sample injection. The standard deviation for replicate
measurements for these standards averaged 0.4‰, with the average offset from their
known values of less than 0.5‰. The isotopic values of FAMEs and acetylated alcohols
were additionally corrected for the introduction of carbon atoms during the
derivatization step.

**2.9. DNA extraction, PCR amplification, Illumina sequencing and data analysis**
DNA was extracted from selected sediment samples of Lake Geneva, where high
methanogenic rates were detected (see below), using FastDNA SPIN Kit for Soils (MP
Biomedicals) following the manufacturer's instructions. A two-step PCR approach was
applied to prepare the library for Illumina sequencing at the Genomics Facility Basel,
as described in detail previously (Su et al., 2020, 2023). PCR was performed using
universal primers 515F-Y and 926R targeting the V4 and V5 regions of the 16S rRNA
gene. These primers cover the majority (~84.7%) of the 16S rRNA gene sequences of
methanogenic archaea and are well-suited for assessing community structures of
methanogens in environmental samples (Table S4). Data were then analyzed with
Phyloseq (McMurdie and Holmes, 2013) in the R environment (R Core Team 2014).
Raw sequence data were deposited at NCBI Short Read Archive under the Bioproject
ID PRJNA736863 with accession numbers from SAMN19667760 to SAMN19667767.

**2.10. Quantitative PCR (qPCR)**
The abundance of methanogens in Geneva sediments was determined by qPCR
with the primer set mcrIRD and 2 µL DNA as template (Lever and Teske, 2015). qPCR
reactions of all DNA samples were performed using the SensiFAST SYBR No-ROX
Kit (Bioline) on a Mic (Magnetic Induction Cycler) real time PCR machine (Bio
Molecular Systems, Inc). An initial denaturing step of 95 °C for 3 min was followed by
40 cycles of 5 s at 95 °C, 10 s at 56 °C, and 22 s at 72 °C. The specificity of the
amplification was assessed by examining the melting curves from 72 °C to 95 °C. The
calibration curve was generated using a serial of 10-fold dilutions of pGEM-T Easy
plasmid DNA (Promega, USA) carrying a single copy of the target gene (mcrIRD-



F/mcrIRD-R). The number of gene copies in plasmid DNA was calculated using the
equation reported previously (Ritalahti et al., 2006).

**3. Results**
**3.1. Sediment geochemistry at the two study sites**

The two sites displayed different hydro- and geochemical characteristics (Fig. 1

and Table S1 and Fig. S1). Dating of the sediments revealed that the deltaic site had a
slightly higher sedimentation rate (0.58 cm yr$^{-1}$) than the profundal site (0.43 cm yr$^{-1}$).
At the deltaic site (DS), two distinct peaks in methane concentration (~7 mM) were
observed at 9 and 23 cm, respectively (Fig. 1A). At the profundal site (PS),
concentrations of CH$_4$ increased with depth and remained relatively constant at ~4 mM
below 15 cm (Fig. 1E). At both sites, the pore water concentrations of dissolved
inorganic carbon (DIC) increased with depth, and considerable amounts accumulated
in the sediment pore water. The depth-integrated DIC concentrations in the top 30 cm
were 1964 mmol m$^{-2}$ at PS and 3490 mmol m$^{-2}$ at DS, however, based on the
sedimentation over the past 50 years, the depth-integrated DIC concentrations were
1302 mmol m$^{-2}$ at PS (21 cm) and 3490 mmol m$^{-2}$ at DS (29 cm). Acetate, as a potential
methanogenic substrate, showed concentration maxima of 25.9 µM (DS) and 3.8 µM
(PS) close to the sediment surface. The methane δ$^{13}$C values decreased with depth in
both deltaic sediments (from -64.9‰ to -72.2‰, Fig. 1B) and profundal sediments
(from -72.7‰ to -74.6‰, Fig. 1F). Overall, the observed methane carbon isotope values
at both sites fall within a range that is typical for biogenic production (Whiticar, 1999).
Vertical profiles of δ$^{13}$C-DIC show similar pattern at the two sites, with the lowest
values of -5.5‰ at 5 cm at DS (Fig. 1B) and -5.2‰ at 3 cm at PS (Fig. 1F). At PS, TOC
concentrations increase slightly with depth, with the mean content almost doubled
compared to DS (Fig. 1C and G, Table S1). With respect to its stable carbon isotope
composition, δ$^{13}$C-TOC increased slightly with depth at PS, from -28‰ in surficial
sediments to -26‰ below 7 cm whereas δ$^{13}$C-TOC values remained relatively constant
at DS throughout the sampled sediment column (~ -26‰). The elemental C/N ratios in
the sediments of DS show a very high variability with depth, and range from 6.7 to
16.8, with a mean value of 10.7 (Fig. 1D and Table S1), while the values at PS are
lower and less variant along the sediment core (7.9 ± 0.3, Fig. 1 H).

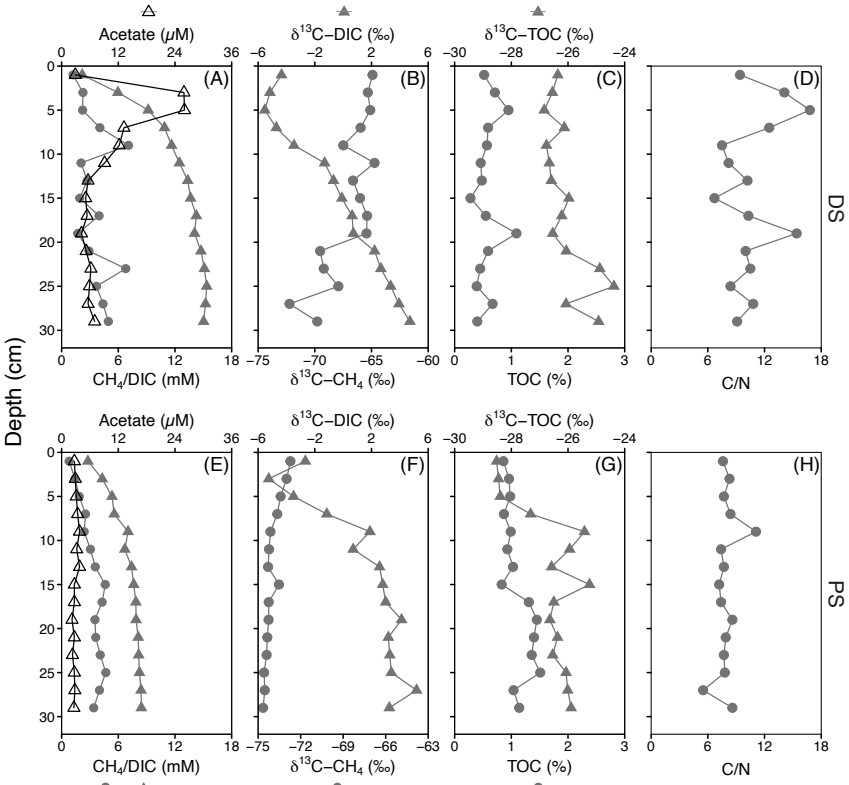


Figure 1. Sediment geochemistry as function of depth at a deltaic site (DS, A-D) and a
profundal site (PS, E-H) in Lake Geneva. (A, E) Profiles of dissolved methane,
dissolved inorganic carbon (DIC) and acetate concentrations. (B, F) Stable carbon
isotopic signatures ($\delta^{13}$C) of CH$_4$ and DIC (in ‰ vs. V-PDB). (C, G) Concentrations of
total organic carbon (TOC, % of dry weight) and its carbon isotopic composition (in ‰
vs. V-PDB). (D, H) Molar ratios of total organic carbon to total nitrogen (C/N).

**3.2. Composition of sedimentary organic matter**

To investigate the molecular composition of sedimentary organic matter at the two

different sites, sediment samples were further analyzed using Py-GC/MS. We identified
a total of 65 individual organic compounds (Table S2), which can be classified into the
following compound groups: carbohydrates, N-compounds, *n*-alkenes, *n*-alkanes,
phenols, lignin oligomers, and (poly)aromatics. Both carbohydrates and N-compounds
were the most abundant organic compound groups among the pyrolysis products, with
no statistical difference between DS and PS (Fig. 2 A and B). At both sites, the





carbohydrates consisted mainly of compounds such as butenal, methylfuraldehyde and
furanone, and N-compounds were dominated by pyrrole, pyridine, methyl-pyrrole and
methyl-pyridine (Table S2), which are indicative of degraded products of carbohydrates
and proteins (Schellekens et al., 2009; Tolu et al., 2017). Most strikingly, sediments at
DS displayed significantly higher relative abundances of lignin ($p < 0.05$) and phenols
($p < 0.01$) compared to profundal sediments (Fig. 3D and E). In addition, we observed
significantly higher relative abundances of $n$-alkenes at PS (Fig. 3C). This group
contains monounsaturated short-chain $n$-$C_{15}$ to $n$-$C_{18}$ (Table S2), which were the most
abundant organic compounds among the $n$-alkenes.

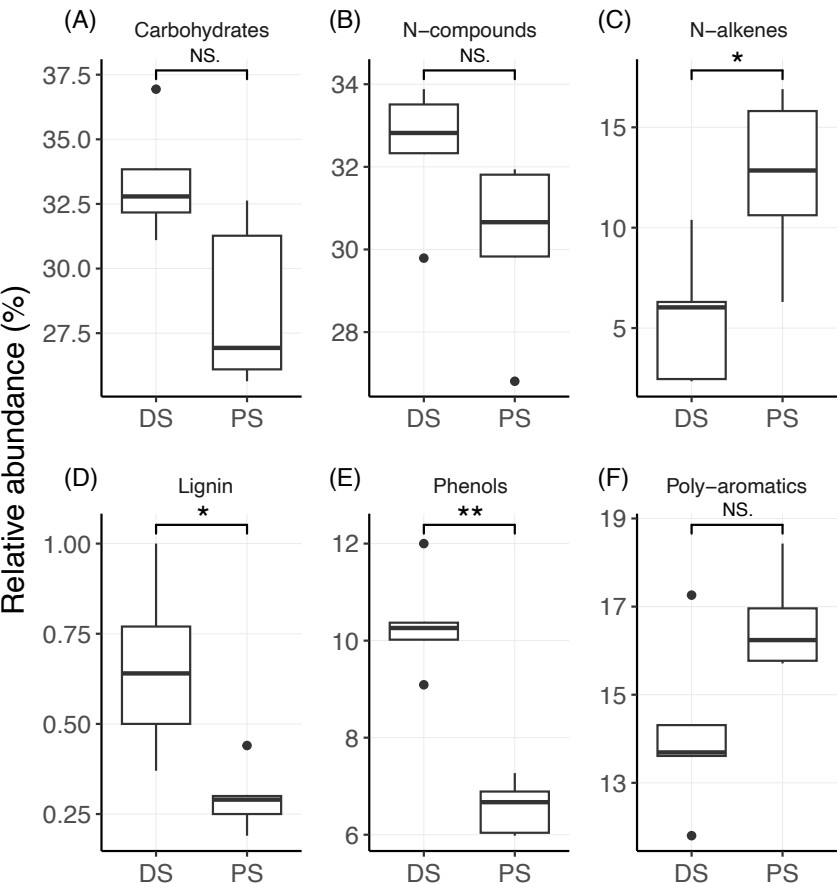


Figure 2. Relative abundances (in %) of different biochemical classes of organic
compounds in sediments at the deltaic (DS) and profundal site (PS) of Lake Geneva.
(A) Carbohydrates, (B) N-compounds, (C) N-alkenes, (D) Lignin, (E) Phenols and (F)



Poly-aromatics. Statistical differences of compound abundances between the two sites
were determined with the Wilcoxon signed-rank test. Significance levels are: ** p<0.01;
* p<0.05; NS. no significant differences between the two sites (p>0.05).

**3.3. Lipid concentration, distribution, and stable carbon isotopic signature**
Typical lipid biomarkers derived from aquatic phytoplankton (e.g., short-chain
fatty acids) and terrestrial plants (e.g., long-chain n-alkanes) were present in both
profundal and deltaic sediments, yet concentrations of some of these biomarkers were
strikingly different between the two sites, as well as between different depths within
the same site (Fig. 3 and Table S3). Among the fatty acids, $n$-$C_{16:0}$ was by far the most
abundant in all measured samples, with a slight increase in $\delta^{13}C$ values with depth at
both sites. At DS, the unsaturated fatty acids $n$-$C_{16:1\omega7}$ was the most abundant
monounsaturated fatty acids with the highest concentration observed at 11cm. At PS,
$n$-$C_{16:1}$ ($\omega7$ and $\omega5$) and $n$-$C_{18:1}$ ($\omega9$ and $\omega7$) decreased in concentration with depth and
were two to four times more abundant in the upper sediment layers (0-2 cm, 4-6 and
10-12 cm) than in the lower parts (18-20 and 28-30 cm). The short-chain fatty acids
were generally depleted in $^{13}C$, with stronger $^{13}C$-depletions observed in surface
sediments, particularly for the unsaturated fatty acids $n$-$C_{16:1\omega5}$ at PS.
In contrast to short-chain fatty acids, $C_{24:0}$ was most abundant among the long-
chain fatty acids (i.e., $C_{24:0}$, $C_{26:0}$ and $C_{28:0}$), with relatively higher concentrations found
in profundal sediments. Most strikingly, an apparent increase in the concentrations of
these long-chain fatty acids was observed in the sediments of PS, with consistently high
concentrations in the lower parts (10-30 cm). Meanwhile, the $\delta^{13}C$ values of these
compounds decreased dramatically with depth. By comparison, concentrations of long-
chain fatty acids at DS were variable at different sediment depths, but their $\delta^{13}C$ values
remained relatively constant (e.g., $C_{28:0}$, Fig. 3B) and showed less variability with depth
compared with PS (e.g., $C_{24:0}$ and $C_{26:0}$). Although the concentrations of long-chain n-
alkanes at DS were twice as high as those at PS, their $\delta^{13}C$ values (particularly for $C_{29}$,
$C_{31}$ and $C_{33}$) were very similar at the two sites (Fig. 3B).



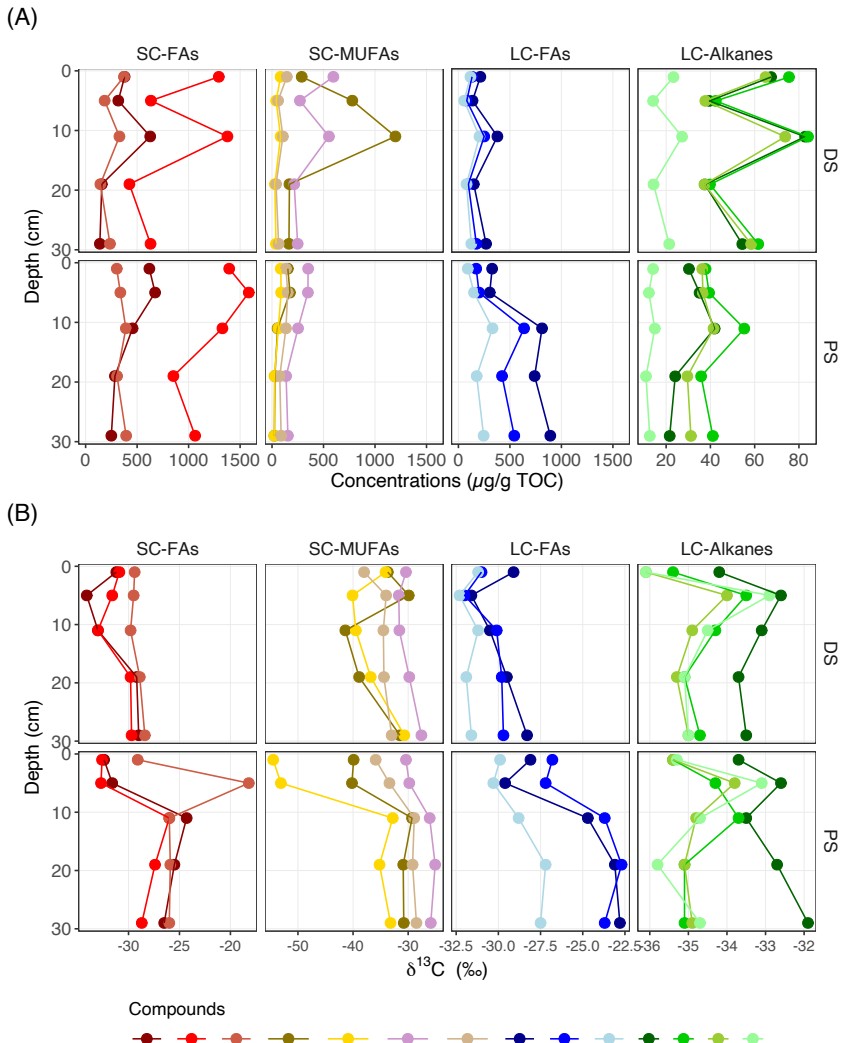

Figure 3. Vertical distribution of specific fatty acids and n-alkanes in deltaic (DS) and profundal (PS) sediments of Lake Geneva. (A) Concentrations (expressed as µg lipid $g^{-1}$ TOC), and (B) compound-specific $\delta^{13}C$ values (in ‰ vs. V-PDB). SC-FAs, short-chain fatty acids ($C_{14:0}+C_{16:0}+C_{18:0}$); SC-MUFAs, short-chain monounsaturated fatty acids ($C_{16:1}$ and $C_{18:1}$); LC-FAs, long-chain fatty acids ($C_{24:0}+C_{26:0}+C_{28:0}$); LC-Alkanes, long-chain n-alkanes ($C_{27}+C_{29}+C_{31}+C_{33}$).

## 3.4. Relative importance of methanogenic pathways



Methane production through $CO_2$ reduction (i.e., hydrogenotrophic
methanogenesis) was observed at both sites and at all depths. At the deltaic site, highest
activity of hydrogenotrophic methanogenesis was found at the lower part of the
sediment core (19-29 cm), while in the profundal sediment, maximum rates of
hydrogenotrophic methanogenesis were detected within the upper sediments at depths
between 7 and 17 cm (Fig. 4B and E, Table S1). In contrast, the rates of methane
production via acetate fermentation (i.e., acetoclastic methanogenesis) were very low,
with maxima of 2.8 and 1.0 nmol $cm^{-3}$ $d^{-1}$ observed at 3 cm (PS) and 7 cm (DS),
respectively. The areal methanogenesis rates by $CO_2$ reduction within the top 30 cm, at
both sites, were two orders of magnitude higher than acetoclastic methanogenesis rates
(Table S1). Acetate oxidation activity was very low at both PS and DS, with the depth-
integrated (0-30 cm) rates of 0.1 and 0.2 mmol $m^{-2}$ $d^{-1}$, respectively (Table S1). At both
sites, the fractionation factor $\alpha_c$, which is approximated based on the difference between
the measured $\delta^{13}$C-DIC and $\delta^{13}$C-$CH_4$ values, increased with depth (Fig. 4F), ranging
from 1.064-1.071 (DS) and 1.073-1.084 (PS).

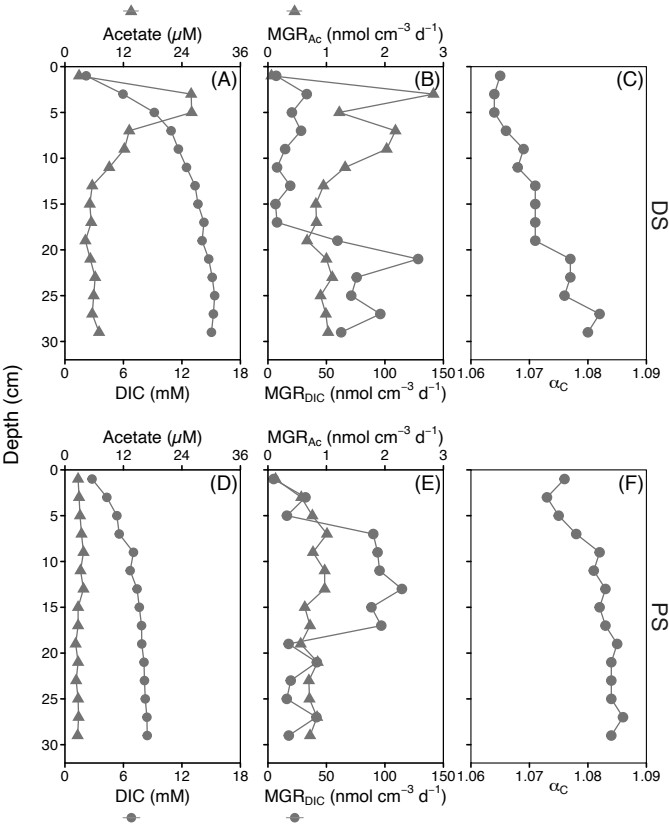

Figure 4. Depth profiles of (A, D) porewater dissolved inorganic carbon (DIC) and acetate, (B, E) $CO_2$ reduction ($MGR_{DIC}$) and acetoclastic ($MGR_{Ac}$) methanogenesis rates, (C, F) apparent fractionation factor $\alpha_C$ in sediments of deltaic (DS) and profundal site (PS) in Lake Geneva. $\alpha_c$ is an approximation of the apparent C-isotope fractionation during methanogenesis, calculated from $\delta^{13}C$-$CH_4$ and $\delta^{13}C$-DIC values, where $\alpha_c =$ $(1000 + \delta^{13}C\text{-DIC})/(1000 + \delta^{13}C\text{-}CH_4)$. Values of $\alpha_c > 1.065$ indicate predominance of $CO_2$ reduction over acticlastic methanogenesis (Conrad, 2005b).

### 3.5. Abundance and diversity of methanogenic archaea

On average, gene copy numbers of the methyl coenzyme M reductase gene (*mcrA*) in the deltaic sediments of Lake Geneva were significantly higher than those in the profundal sediments ($p < 0.05$; Fig. 5A). At both sites, *Methanoregula* and *Methanothrix* dominated the methanogenic guild, followed by *Methanosarcina* and *Methanobacterium* (Fig. 5B). Within the methanogenic community, the mean relative



abundances of *Methanothrix* and *Methanosarcina* were both higher at DS than those at
PS. By comparison, *Methanoregula* was the most abundant genus in the methanogenic
community at PS. This matches recent studies on other lakes in Switzerland, in which
Methanoregula was found to dominate methanogenic communities in the sediments
(Bartosiewicz et al., 2024; Meier et al., 2024). Both *Methanoregula*, *Methanobacterium*
and *Methanosphaerula* use $H_2/CO_2$ as the main substrates for methanogenesis (Oren,
2014a). While *Methanothrix* have long been considered to use exclusively acetate as
substrate for methanogenesis (Jetten et al., 1992; Smith and Ingram-Smith, 2007), there
is now evidence that some species can also perform $CO_2$ reduction via direct
interspecies electron transfer (DIET) (Rotaru et al., 2014; Zhou et al., 2023). Also,
*Methanosarcina* can utilize a broader range of substrates, including $CO_2$ reduction with
$H_2$ or via DIET, acetate, as well as some methylated compounds (Oren, 2014b).

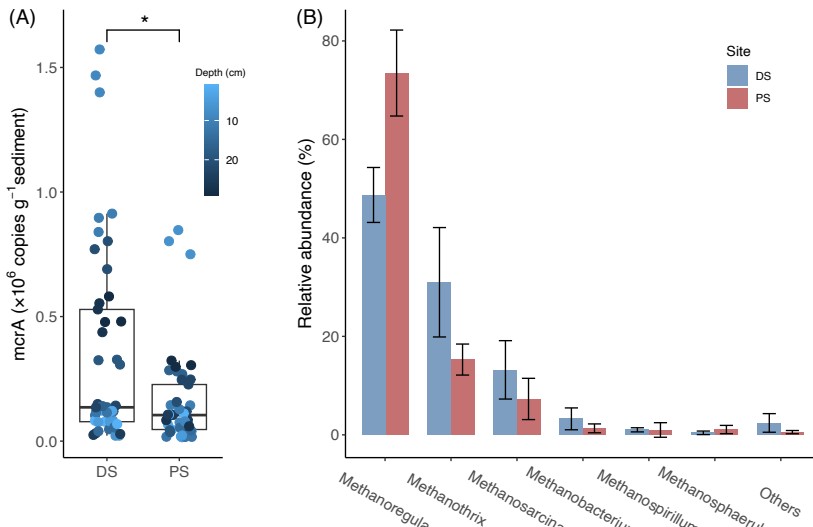


Figure 5. Abundance and diversity of methanogens at the deltaic (DS) and the profundal
site (PS) in Lake Geneva. (A) Absolute abundances of the *mcrA* gene encoding the α-
subunit of the methyl-coenzyme M reductase. Statistical difference between the two
sites was determined with the Wilcoxon signed-rank test and asterisk denotes
significance level (*: $p<0.05$). (B) Mean relative abundances of different methanogenic
groups (% of the total methanogens). Data are based on read abundances of 16S rRNA
gene sequences.





**3.6. Correlation analysis between geochemical parameters, methane production rates, and microbial communities**

The alpha diversity measures indicate that the community structure of microorganisms in the sediments at DS were more diverse than at PS (Fig. S2). Principal coordinate analysis (PCoA) revealed significantly different microbial community compositions at the two investigated sites, as indicated by the clear separation of the data by the first principal coordinates, explaining 50.8% and 44.5% of the observed variance for archaea and bacteria, respectively (Fig. S3). For both archaeal and bacterial communities, PCoA plots show a very close aggregation of the deep sediment samples at PS (17, 23 and 29 cm), indicating that their community structures are highly similar. Conversely, there is quite some variance among the samples at the more dynamic deltaic site DS, with its variable deposition history. The measured rates of both $CO_2$ reduction and acetoclastic methanogenesis tend to be higher in sediment samples harboring a more diverse microbial community (Fig. S4), but the relation is weak and particularly for acetoclastic methanogenesis not significant. Pearson correlation analysis between methane production rates and environmental parameters show that methane production rates from both $MGR_{DIC}$ and $MGR_{Ac}$ was positively but not significantly correlated with the concentration of short-chain n-alkanes (correlation coefficients of 0.46 and 0.51 for $MGR_{DIC}$ and $MGR_{Ac}$, respectively, Fig. S5). $\delta^{13}C$-$CH_4$ values showed a positive correlation with the abundance of carbohydrates, lignin, and long-chain n-alkanes, and significant negative correlation with the abundance of long-chain fatty acids and total organic carbon. C/N ratios were negatively correlated with the concentrations of both short-chain and long-chain fatty acids (Fig. S5).

**4. Discussion**

Our results indicate a dominance of methane production by $CO_2$ reduction in both profundal and deltaic sediments of Lake Geneva. This inference is supported by radiotracer measurements, the observed apparent fractionation factors, and methanogenic community analyses, which revealed members of $CO_2$-reducing *Methanoregula* as the dominant group of methanogens. Thus, $CO_2$ reduction was observed as the primary MGR process both in sediments at PS, which predominantly contained diagenetically altered phytoplankton-derived OC, and sediments at DS





characterized by variable sources of aquatic and terrestrial OC. We conclude, therefore,
that the dominant pathway of methanogenesis does not primarily depend on the
chemical composition of sedimentary organic matter. Other factors that affect
production rates of different electron donors (e.g., $H_2$, acetate) could play a more
important role, and will be discussed below.

**4.1. Difference in organic carbon sources and diagenetic alteration**

The deltaic sediments in this study displayed a strikingly large range of C/N ratios

(6.7-16.8), whereas C/N ratios of profundal samples remained relatively constant
throughout the sediment core (7.9 ± 0.3). Fresh organic matter from lacustrine
phytoplankton, which is protein- rich (Parsons et al., 1961), typically has low C/N ratios
of 6-9 (Meyers and Ishiwatari, 1993). In contrast, bulk sediment containing large
portions of terrestrial vascular plants often displays much higher C/N ratios, sometimes
at least three times greater, due to its high content of high-carbon structural components
and refractory organics such as lignin (Hedges et al., 1986). A comparison of carbon
isotope and bulk C/N values to data from previous studies (Lamb et al., 2006)
confirmed the presence of both aquatic and terrestrial sources in the deltaic sediments,
and the dominance of organic matter of autochthonous origin (e.g., algal biomass) at
PS (Fig. 6). At DS, variable microbial community structures and significant changes in
C/N ratios with depth reflect a highly variable depositional history, indicating differing
origins of sedimentary organic matter. Specifically, peak C/N ratios of 16.8 at 5 cm and
15.4 at 19 cm were likely derived from terrestrial organic matter in these intercalated
layers, while relatively lower ratios (~10) at other depths suggest a mixture of aquatic
and terrestrial sources. Even lower values (~7) in some deltaic sediment layers may
indicate predominantly autochthonous deposition of phytoplankton particles.



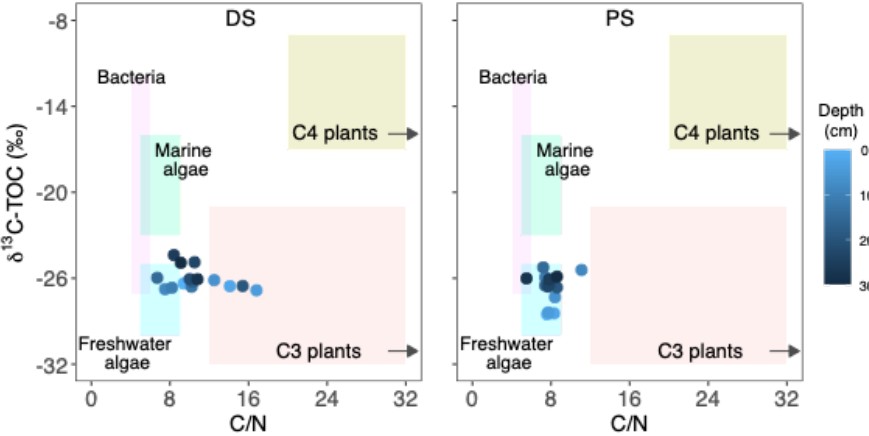


Figure 6. Comparison of $\delta^{13}C$ and bulk C/N values of Lake Geneva sediments from
the deltaic (DS, left) and profundal site (PS, right) to elemental and isotopic indicators
of bulk organic matter produced by bacteria, marine and freshwater algae, as well as
terrestrial C3 and C4 plants (Lamb et al., 2006).

590   Indeed, at both sites, sediments contained OC from aquatic biomass, as verified by

the short-chain fatty acids, indicative of freshwater algae (Cranwell, 1976), and/or in
situ production by bacteria (Volkman et al., 1998). However, contrary to the findings
from the combined carbon isotope and bulk C/N ratios, both lignin/phenols (Fig. 2) and
long-chain n-alkanes (Fig. 3A) derived from terrestrial plants were detected in
profundal sediments, although the concentrations of these compounds were lower at PS
than at DS. Long-chain fatty acids (i.e., $C_{24:0}$ and $C_{26:0}$), which are typically assumed to
originate from terrestrial plant waxes (Cranwell, 1974), make up a proportionally more
important fraction of lipids in deeper sediment layers, particularly below 5 cm depth.
However, the distinct carbon isotope compositions of these long-chain fatty acids
between the two sites (Fig. 3B and Table S3, $\delta^{13}C$ on average ~7‰ lower at DS than at
PS) suggest different OC sources.

602   Indeed, Chikaraishi et al. (2004) reported that long-chain fatty acids (i.e., $C_{24:0}$ and

$C_{26:0}$) have $\delta^{13}C$ values of -36.3±2.6‰ for a variety of terrestrial vascular plants,
whereas the $\delta^{13}C$ for aquatic plants is significantly less negative (-25.5±0.9‰). At PS,
$\delta^{13}C$ values for $C_{24:0}$ and $C_{26:0}$ fatty acids (below 5 cm depth) were close to those
indicative of freshwater aquatic plants (Chikaraishi et al., 2004). Previous studies have
shown that both aquatic macrophytes and microalgae may represent potential sources



of long-chain fatty acids (Ficken et al., 2000; Volkman et al., 1998). However, it seems
somewhat contradictory to conclude that long-chain fatty acids were primarily derived
from aquatic vascular plants, simply because the observed C/N ratios at PS were too
low. Indeed, microalgae such as diatoms were estimated to contribute from 30-80% of
the $C_{24:0}$ to $C_{28:0}$ fatty acid pool in intertidal sandy sediment (Volkman et al., 1980).
Thus, the long-chain fatty acids observed in the profundal sediments may also have
been derived from microalgae (Volkman et al., 1998). This, in turn, could explain why
TOC was more depleted in 13C in the surface sediments at PS, which does not
necessarily indicate the origins of terrestrial plants (Cloern et al., 2002).
The carbon isotopic composition of long-chain n-alkanes (i.e., $C_{27-31}$),
characteristic for land-derived organic matter (Chikaraishi et al., 2004), was almost
identical between PS and DS (Fig. 3B, ranging from -35‰ to -33‰), suggesting a
similar terrestrial source. At DS, if long-chain n-alkanes and long-chain fatty acids were
derived from the same terrestrial source, they should have similar C-isotopic values.
However, the slightly lower $\delta^{13}C$ values of long-chain fatty acids at DS (on average, -
30‰) imply a mixture of both aquatic and terrestrial sources, which is consistent with
the C/N ratios observed at this site.
Turning to other OM biomarkers, phytol concentrations were considerably higher
at PS (with no clear depth trend) compared to DS, whereas no clear difference in
cholesterol concentrations was observed between the sites (with depth-dependent
concentration decrease at both sites). For brassicasterol, there seems to be a clear depth-
related decrease at both sites, with slightly higher concentrations in surface sediments
(0-2 and 4-6 cm) at PS compared to DS. Phytol, a side chain of chlorophyll, can
originate from both phytoplankton and terrestrial plants (Shi et al., 2001). Due to its
rapid degradation under intense-light and oxic conditions in terrestrial environments,
terrestrially derived phytol (or chlorophyll) is generally of minor importance in aquatic
sediments (Meyers and Takeuchi, 1981), so it can be assumed that most of the
sedimentary phytol is derived from aquatic sources (Ladd et al., 2018). As for
cholestanol and brassicasterol (Table S3), the former can originate from both
zooplankton and phytoplankton (Bechtel and Schubert, 2009), while the latter is a lipid
biomarker mainly derived from phytoplankton in aquatic sediments (Volkman, 1986).
Hence, based on at least two of the biomarkers presented here, we have putative
evidence for a shift towards more autochthonous OM at the profundal site, as could be
expected.




### 4.2. Methanogenesis is mainly driven by $CO_2$ reduction

Our rate measurements using trace [14]C-labelled substrates clearly showed that $CO_2$
reduction played a dominant role in methane formation in both the PS and DS
sediments, compared to acetoclastic methanogenesis. As we did not measure rates of
methylotrophic methanogenesis, we are unable to determine the relative importance of
this pathway. However, since we neither detected common methylated compounds
such as methanol in the sediment porewater, nor methanogens that exclusively perform
methylotrophic methanogenesis at any of the studied sites, we argue that this pathway
is likely of a minor importance with regards to its the contribution to the overall
methane production at these sites. Indeed, $H_2/CO_2$ and acetate are usually the main
substrates in freshwater environments (Lyu et al., 2018), and to date, no direct evidence
for the occurrence of methylotrophic methanogenesis has been documented for lake
sediments. Methylotrophic methanogenesis has been shown to be an important pathway
mainly in specific marine systems, where methanogens can utilize methylated
compounds (e.g., methanol and trimethylamine) as non-competitive substrates for
methane production (Xiao et al., 2018; Xu et al., 2021; Zhuang et al., 2018).
The finding that $CO_2$ reduction pathway dominated methane production
throughout the sediment cores at both sites was further supported by the observed
apparent fractionation factor $\alpha_c$ (DS: $1.071 \pm 0.001$ and PS: $1.081 \pm 0.001$; Fig. 4C, F
and Table S1) (Conrad, 2005). Rates for $CO_2$ reduction in Lake Geneva were similar at
the two studied sites, but were much higher than those previously reported in other
lakes at similar depths (Kuivila et al., 1989). The observed low rates for acetoclastic
methanogenesis on the other hand were comparable to those in other lake sediments
(Kuivila et al., 1989; Schulz and Conrad, 1996). Methane production via acetate
fermentation was traditionally believed to play a more important role than via $CO_2$
reduction in lake sediments (Whiticar, 1999; Whiticar et al., 1986). However, more
recent evidence seems to contradict this paradigm, demonstrating that
hydrogenotrophic methanogenesis can indeed be a much more significant methane-
producing biogeochemical pathway than acetoclastic methanogenesis (Blair et al.,
2018; Conrad et al., 2011; Meier et al., 2024). The predominance of methane formation
via $CO_2$ reduction over acetoclastic methanogenesis is an important finding, which, in
Lake Geneva, appears to apply to sediments across different sedimentary depositional
regimes.



At both sites, *Methanoregula* was the most abundant methanogenic genus,
particularly in deep sediments where the highest $CO_2$ reduction rates were observed.
This genus, commonly found in freshwater lakes (Bartosiewicz et al., 2024; Berberich
et al., 2020; Meier et al., 2024), is known to perform hydrogenotrophic methanogenesis.
*Methanothrix*, the second most abundant methanogen, primarily performs acetoclastic
methanogenosis, although some species within this cluster can also produce methane
via $CO_2$ reduction (Rotaru et al., 2014). At PS, *Methanoregula* dominated at all
investigated depths, in accordance with high rates of autotrophic methanogenesis.
While acetoclastic methanogenesis played only a minor role in total methane
production, higher rates were observed for deltaic sediments, with lower apparent
fractionation factor. This observation is consistent with higher concentrations of acetate
and a higher relative abundance of acetate-comsuming *Methanothrix*, suggesting that
the pathway of acetoclastic methanogenesis was influenced by both the availability of
acetate and the functional methanogenic population.
Although overall methane production rates were similar at both sites, the qPCR
data in hand indicated that bulk methane production was not primarily/directly
controlled by (i.e., proportional to) the cell abundance of methanogens. Instead, it was
likely regulated by other environmental factors, such as limited substrate
availability/supply (i.e., $H_2$ and acetate), stemming from rates of hydrolysis (Kristensen
et al., 1995) and/or fermentation rates (Valentine et al., 1994) within the sediments.
Methanogens depend on syntrophic and other heterotrophic bacteria, which may
significantly influence methane production rates (Beulig et al., 2018; Liu and Whitman,
2008).

In Lake Geneva sediments, rates of both $CO_2$ reduction and acetoclastic
methanogenesis increased with microbial Shannon diversity. More diverse microbial
communities tend to possess more diverse organic matter degradation capacities,
resulting in higher production of acetate and $H_2/CO_2$, which in turn promotes overall
methane production (Conrad, 2020). However, our microbial abundance and diversity
data were insufficient to fully characterize the carbon metabolism of microorganisms
involved in $H_2$ and acetate production, which could directly impact $CO_2$ reduction and
acetoclastic methanogenesis and potentially explain the observed methane production
differences. The very low acetate oxidation rates at both sites suggest that differences
in methane generation via $MGR_{DIC}$ versus $MGR_{Ac}$ may be attributed to the relative
availability of methane precursors available (i.e., $H_2/CO_2$ versus acetate) (Capone and



Kiene, 1988). The acetate concentrations were low (below 26 µM) at both sites, but
well within the range of what has been measured in profundal sediment of Lake
Constance, where acetoclastic methanogenesis was dominant (Schulz and Conrad,
1996). Hence, the consistently low rates of acetoclastic methanogenesis throughout the
sediment cores at both sites likely resulted from low turnover rates of acetate during
organic matter degradation.

**4.3. Effect of organic carbon characteristics on methanogenesis**
Despite the differences in OC sources and quality between the two sites, the overall
methane production rates were comparable (Fig. 4B, E and Table S1). However, the
sedimentary zones of maximum methane production and the depth-dependent rates of
both metabolic modes of methanogenesis varied. This likely reflects differences in OC
quality and degradation pathways, which influence the balance between
remineralization to $H_2/CO_2$ versus acetate (Conrad, 2020), resulting in spatial
variability in sedimentary methane production.
Potential methane production rates have been shown to correlate positively with
the quantity of sediment organic matter (Berberich et al., 2020). However, at both the
DS and PS in Lake Geneva, the sediment layers of maximum rates of $CO_2$ reduction
did not coincide with higher TOC contents. Instead, combined rate and isotopic
measurements revealed that methanogenesis rates via $CO_2$ reduction corresponded to
less $^{13}$C-depleted bulk organic carbon (Fig. 1 and 4). This suggests that the source and
microbial decomposition of organic matter, rather than its quantity, primarily
influenced the methanogenic activity via the $CO_2$-reduction pathway.
Methane production rates in lake sediments are generally low across various
latitudes but can increase significantly with the addition of fresh organic carbon
(Bartosiewicz et al., 2024; Schwarz et al., 2008; West et al., 2012). Indeed, high lipid
contents in phytoplankton biomass have been shown to enhance methane production in
both engineered systems and lake sediments (West et al., 2015; Zhao et al., 2014).
Additionally, existing evidence suggest that terrestrial OC can stimulate methane
ebullition in reservoirs (DelSontro et al., 2011; Sobek et al., 2012). Our findings from
rate measurement and organic carbon composition agree with earlier studies, indicating
that both aquatic and terrestrial OC can significantly contribute to methane production
(Berberich et al., 2020; Grasset et al., 2018). It has also been suggested that the
availability of easily degradable organic matter controls both methanogenic pathways



and methanogenic archaeal communities (Liu et al., 2017). However, the influence of
organic matter source or composition on the relative importance of methane production
pathways and their corresponding rates remains poorly constrained and has rarely been
explored. Within the zone of high hydrogenotrophic activity at PS, TOC became less
depleted in $^{13}$C, a trend also observed, though less pronounced, in the deltaic sediments.
The $\delta^{13}$C shift of TOC within the most active zone at PS likely resulted from the
selective loss of isotopically light OM fractions (Lehmann et al., 2002b; Meyers, 1994),
or the changes in organic matter sources, as suggested by variations in both short-chain
and long-chain fatty acids indicative of autochthonous origins.

Contrary to previous studies demonstrating that algal deposition stimulated the

activity of acetoclastic methanogens (Schulz and Conrad, 1995; Schwarz et al., 2008),
our results revealed higher acetate concentrations in tandem with higher acetoclastic
methanogenesis occurring in surface sediments of deltaic site, where high C/N ratios
indicated terrestrial inputs. This suggests that allochthonous OC may play a role in
acetate production and acetoclastic methanogenesis. However, the apparently much
lower overall acetoclastic rates compared $CO_2$ reduction were likely due to the active
acetate consumption via syntrophic oxidation (Conrad et al., 2020). At PS,
autochthonous organic matter was the dominant OC source to the sediments, and the
$CH_4$ produced likely resulted from the decomposition of buried algal detritus (Fig. 6).
This interpretation is also supported by the 2-fold higher brassicasterol, the 3-fold
higher phytol, and 3-4-fold higher long chain fatty acids concentrations (with elevated
$\delta^{13}$C values) at PS compared to DS. These biomarkers, in combination with lower C/N
ratios, are indicative of algal material as the dominant organic contributor to early
diagenetic methane production.

While the dominant sedimentary organic matter source (i.e., the partitioning

between autochthonous versus allochthonous OM inputs to the sediments) varied
between the two studied sites, the similar overall methane production rates imply that
methanogenesis occurred ubiquitously in these lacustrine sediments, independent of the
OC sources, their apparent susceptibility to organic matter degradation, and/or their
overall diagenetic state. This finding aligns with a recent study across the river/deltaic-
pelagic continuum in a reservoir system in southwest Ohio (Berberich et al., 2020), and
holds important implications for understanding the lacustrine methane cycle. On the
one hand, eutrophication in lakes and other aquatic systems often increases the supply
of autochthonous organic carbon, thereby enhancing methanogenesis and leading to





increased methane emissions (Davidson et al., 2018). On the other hand, the potential
of terrestrial organic matter to contribute to lacustrine methane production can be
significantly impacted by anthropogenic activities, such as dam construction (Li et al.,
2023), or as a result of deforestation and soil erosion, which alter sediment inputs into
the lake (Bélanger et al., 2017). Difference in the source and lability of organic carbon
in the lacustrine organic matter pool may affect sediment methane production in distinct
ways (Grasset et al., 2018; Stibal et al., 2012). Our study demonstrates that diverse OC
sources and varying diagenetic states in both profundal and deltaic sediments support
substantial rates of methane formation through $CO_2$ reduction. However, identifying
the exact fractions of organic matter that were degraded and fueled methane production
remains challenging. While acetoclastic methanogenesis appears to be relevant to the
break-down and decomposition of terrestrial organic matter that favors acetate
production and accumulation in deltaic sediments, the sources of organic carbon driving
the more dominant $CO_2$-reducing methanogenesis in the studied sediments remains
unclear. The key environmental factors controlling the organic matter degradation in
methanogenic sediments remain unresolved and require further investigation.
Understanding these mechanisms is essential for predicting how changes in organic
carbon inputs and sediment dynamics may influence methane production and emissions
from lacustrine environments.

## 5. Conclusions

Our study demonstrates that both profundal and deltaic sediments of Lake Geneva
are significant sources of methane, despite clear differences in origins and compositions
of organic matter. The profundal site is dominated by aquatic OM, while the deltaic site
features a more variable mix of OM sources, including substantial terrestrial
contributions at certain depths. Methane production at both sites is overwhelmingly
driven by $CO_2$ reduction, which accounted for over 95% of total methane production
and was most probably mediated by *Methanoregula*. While phylogenetic data suggest
a link between methanogen communities and methanogenetic pathways, the broader
microbial community dynamics, particularly those involved in $H_2$ and acetate
production, remain insufficiently understood.
At the profundal site, methane production is mainly associated with the
decomposition of aquatic organic matter, as terrestrial OC is less abundant. At the



deltaic site, acetoclastic methanogenesis appears linked to the higher terrestrial organic
matter inputs. Nevertheless, the overall dominance of methanogenesis via $CO_2$
reduction at both sites suggests that depositional regime, as well as OM source and
composition are not the primary determinants of the prevailing methane producing
pathway. Instead, factors such as the metabolic interactions within microbial
communities (e.g., with syntrophic partner organisms), particularly the balance of $H_2$
and acetate production, may play a larger role. Future research should focus on
unraveling these interactions to better predict if, and how, changes in OM inputs and
sediment dynamics may impact methane emissions from lacustrine environments.



**Data availability statement**

Raw reads of the 16S rRNA sequencing data are available on NCBI GenBank with BioProject ID PRJNA736863, under accession number SRR14794332- SRR14794339.

**Author contributions**

CJS, GS and MAL conceived the research. CJS acquired funding for this study. GS performed field work, lab experiments and data analyses. JT assisted with Pyrolysis-GC data analysis. CG and MAL measured porewater volatile fatty acids and analyzed the data. JZ and MFL helped with rate measurements and molecular analyses. GS wrote the original draft of the manuscript. All authors revised and approved the submitted version.

**Acknowledgements**

This research was supported by Eawag internal funds. We thank Alois Zwyssig, Sandra Schmid and Cameron M. Callbeck for assistance with field sample collection. We also thank Serge Robert for laboratory assistance including lipid sample preparation and laboratory analyses and for his help in elemental analysis of samples. Patrick Kathriner is acknowledged for laboratory support. We are also grateful to Thomas Kuhn at the University of Basel for laboratory support with the radioisotope measurements.

**Competing interests**

At least one of the (co-)authors is a member of the editorial board of Biogeosciences.



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
