# Peer review of "Methanogenesis by CO2 reduction dominates lake sediments"

_EGUsphere, 2025_

## Author Comment (AC1)

**Response (in bold) to comments by the Referee#1**

This manuscript presents data on the regulation of methanogenesis in two different sediments of Lake Geneva. The authors measured depth profiles of acetoclastic and hydrogenotrophic methanogenesis in sediments from a deep profundal and a shallower deltaic site and related these rates to sediment properties, biomarker abundance and isotope signatures, and the abundance and diversity of the methanogenic community. Despite significant differences in organic matter quantity and quality, and methanogen abundance, the authors found similar methane production rates, similar methanogen community composition and a clear dominance of hydrogenotrophic methanogenesis at both sites.

The manuscript addresses an important research question, presents very interesting and partly surprising data, and the results of the different approaches used provide new insights into the regulation of methanogenesis in lake sediments. The manuscript is well written and the conclusions are supported by the data. I have only a few minor suggestions to improve this very nice and readable manuscript. First, I found the discussion too long. It could be shortened considerably, especially by omitting statements about obvious facts (see examples below). Also, most of the conclusion is a summary of the results. Therefore, the last part of the conclusion can be revised (see below) and combined with the discussion.

**Response: We thank the reviewer for this positive assessment.**

Specific comments:

L45f: Is there another source of methane in lake sediments besides anaerobic decomposition of organic matter?

**Response: Anaerobic decomposition of organic matter is generally the predominant source of methane in lake sediments. However, recent evidence has suggested that many other organisms such as fungi, algae, cyanobacteria and bacteria can produce methane in the presence of oxygen. For example, Ernst et al demonstrated that methane formation by Bacillus subtilis and Escherichia coli was triggered by free iron and reactive oxygen species (Ernst, L. and others. 2022. Methane formation driven by reactive oxygen species across all living organisms. Nature 603: 482–487. doi:10.1038/s41586-022-04511-9). We have no reason to assume, however, that these processes play any significant role in the studied sediments. Now, in the revised manuscript, we write "Although recent**

**evidence suggests that various living organisms can produce methane under oxic conditions (Ernst et al., 2022), most CH4 in lakes is still produced primarily during the anaerobic decomposition of organic carbon (OC) in sediments,....” (Line 45-48)**

L77ff: This is a very long sentence that I had to read several times to understand. Perhaps it should be split into two parts.

**Response: Thank you for pointing this out. We have revised the sentence for clarity and split it into two shorter sentences (Now at lines 78–84)**

L98: There are also terrestrial C4 plants such as maize

**Response: Yes, terrestrial C4 plants such as maize do indeed use the Calvin-Benson cycle (also known as the C3 pathway), but only after initially concentrating CO2 through the C4 pathway, this $CO_2$-concentrating mechanism is what leads to their characteristically higher (less negative) $\delta^{13}C$ values compared to C3 plants. However, land cover in the Lake Geneva catchment is predominantly composed of temperate forests, grasslands, and pastures, which are overwhelmingly dominated by C3 vegetation. While C4 crops like maize may be cultivated in parts of the region, their spatial extent is limited, and there is currently no evidence suggesting that they contribute significantly to the terrestrial organic matter input into the lake. To avoid any imprecision, we refer specifically to terrestrial C3 plants rather than terrestrial plants in general.**

L273ff: What components have been quantified with this method?

**Response: We provided details regarding the quantified components in the Supplementary Information (Table S2). Briefly, we quantified organic compounds such as n-alkanes, n-alkenes, carbohydrates, N compounds, Phenols and lignin oligomers, and (poly)aromatics.**

L291ff: Please explain which depths were chosen and why.

**Response: The depth information (1, 5, 11, 19, and 29 cm) has now been added to clarify our rationale for selecting these specific intervals. The surface sediment (1 cm) represents the oxic-to-suboxic zone, characterized by fresh organic matter input and active early diagenetic processes. Depths of 5 cm and 19 cm were chosen because they correspond to the highest total organic carbon (TOC) concentrations at the two study sites. Depths 11 cm and 29 cm were selected based on elevated methanogenesis rates. Together, these depths also provide a representative vertical distribution across the sediment profile. In the manuscript, now it reads: "At both sites, sediment samples from five depths (1, 5, 11, 19, and 29 cm) were selected based on redox conditions, organic carbon content, and methanogenesis rates, then lyophilized and homogenized for subsequent lipid extraction." (Lines 302-304)**

L350: 'below' I could not find any information about methanogenic rates or the depths at which samples were taken. Please explain what depths were sampled for DNA extraction.

**Response: We have clarified the exact depths at which DNA samples were taken. Now it writes "DNA was extracted from selected sediment samples of Lake Geneva, where both high and low methanogenic activity were detected (DS: 11, 19, 21 and 27 cm; PS: 5, 11, 19, and 29 cm), using FastDNA SPIN Kit for Soils (MP Biomedicals) following the manufacturer's instructions."**

L374: The section on statistics is missing.

**Response: We thank the reviewer for pointing this out. A new subsection detailing the statistical analyses has been added at the end of section 2.10. *Quantitative PCR (qPCR)*; specifically "We further compared the absolute gene copy numbers of mcrA between the two sites using the Wilcoxon signed-rank test ('wilcox.test'), as implemented in the R package 'stats' (R Core Team, 2024).".**

L484: The difference between the d13C values of CH4 and TIC is about 60, please explain in the M&M section how the fractionation factor was approximated.

**Response: We have now included an explanation in the M&M section (2.4. *Pore water methane and nutrient analyses*) of how the isotope effect was calculated (Lines 245–249).**

L505ff: The second part of this paragraph should be in the discussion.

**Response: We agree with the reviewer's suggestion and have moved the latter part of the paragraph to the Discussion section. We also made minor deletions to enhance conciseness (Lines 686–689).**

L681: typo: methanogenesis

**Response: Corrected.**

L710f: What is meant by "low" in this context? The concentration of acetate seems to be higher than the concentration of any other low molecular weight fatty acid at the DS site. Also, it's difficult to infer the importance of an intermediate from its concentrations. "Low" concentrations may be due to high rates of degradation or low rates of production.

**Response: We agree that "low" was imprecise. The sentence has been rephrased and now it reads "The acetate concentrations were below 26 µM at both sites". This is now at Line 743 in the revised manuscript.**

L739ff: This is a rather obvious statement that is probably not needed.

**Response: We have removed this statement in the revised manuscript to improve conciseness.**

L782ff: This is a very general statement. However, it could not be supported by the current study.

**Response: Thank you for pointing this out. We agree that the statement was too broad and not directly supported by our data. To avoid overinterpretation and ensure consistency with our findings, we have removed it.**

L815ff: This is obvious since H2 and acetate are the main substrates of methanogenesis, not sedimentary organic matter.

**Response: We agree with the reviewer. However, sedimentary organic matter plays a critical role as the precursor for the production of $H_2$ and acetate, the main substrates for methanogenesis. In the manuscript, we revised the sentence as follows to incorporate the role of sedimentary organic matter: "Instead, other environmental factors such as substrate availability or microbial community dynamics may influence methane production, though the specific mechanisms remain uncertain. For example, sedimentary organic matter plays a critical role as the precursor for the production of $H_2$ and acetate, the main substrates for methanogenesis." (Lines 810-814)**

Fig. 1 and Fig. 4: The acetate and DIC concentration profiles should be shown only once. In addition, the lines and symbols of different data should be changed to better differentiate between different parameters.

**Response: We have removed the profiles for acetate and DIC (i.e., panels A and D) from the original Fig. 4. In addition, we have also adjusted the line styles and symbols in both Fig. 1 and Fig. 4 for clarity and consistency.**

Fig. 5: Please align the methanogen groups with the tics of the respective columns.

**Response: We have corrected the alignment of methanogen groups with the axis ticks in Fig. 5.**

---

## Author Comment (AC2)

**Response (in bold) to comments by the Referee#2**

This study investigated methane production pathways at two profundal and deltaic sites in sediment of Lake Geneva and sought to examine how the sources and compositions of organic carbon impact methanogenic potential. The authors measured stable isotopes of methane, methanogenesis rates with radiotracers, and methanogen communities and concluded that hydrogenotrophic methanogenesis was the dominant methanogenic pathway at both sites. They analyzed the concentrations and carbon isotopic compositions of long-chain fatty acids to constrain the source of organic carbon, but did not observe a clear effect of OC source on methane production. The methods were all sound, the experiments were well designed and the results were interpreted correctly. Overall, this study is important to help understand the pathways and controls of methane production in lacustrine sediments. I really appreciate this work and only have some minor comments before acceptance.

**Response: We thank the reviewer for this positive assessment.**

1. Line 177-178. Please double check labeled position of carbon in acetate. As far as I know, 2-$^{14}$C-labeled acetate is $^{14}$C in the methyl group rather than carboxyl group. This could be important for the interpretation of the results, as the methane carbon is derived from the methyl group during the acetoclastic methanogenesis.

   **Response: We thank the reviewer for catching this. We have corrected the description of the 2-$^{14}$C-acetate labeling to clarify that the $^{14}$C label is in the methyl group, which is the carbon converted to methane (Line 178).**

2. Line 394-400: I am curious why DIC concentrations were much higher at the low TOC site of DS?

   **Response: This is an interesting point. Possible explanations may include differences in carbonate buffering capacity, respiration pathways, or porewater exchange. However, we decided not to speculate about this aspect further.**

3. Line 560-562: The authors mentioned the production rates of $H_2$ or acetate) could impact methanogenic pathways, so I wonder if you measured hydrogen concentrations in the original sediments.

**Response: We did not measure hydrogen concentrations in this study. However, we note that while hydrogen concentration measurements can provide some insights, they do not necessarily reflect production or consumption rates. Accurate assessments of hydrogen dynamics would require dedicated incubation experiments, which were beyond the scope of this study.**

4. Line 672-675: Since a number of previous studies observed the dominance of hydrogenotrophic methanogenesis in lake sediment, this study further confirmed this conclusion with more concrete evidence, but this is not a new finding. So I would suggest to rephrase the text here.

**Response: We agree that this observation supports, rather than introduces, a novel finding. We have rephrased the text accordingly: "While the dominance of hydrogenotrophic ($CO_2$ reduction) methanogenesis in lake sediments has been reported previously, our results reinforce this conclusion with consistent geochemical and isotopic evidence across contrasting sedimentary regimes in Lake Geneva." (Line 669-672).**

5. Line 696-698: This sentence came lack of context. Please clarify.

**Response: We have revised this sentence ("Since methanogens rely on syntrophic and other heterotrophic bacteria to generate these key substrates, the activity and composition of the broader microbial community may play a significant role in shaping methane production rates", Lines 695-698), and integrated it with the following paragraph to provide better context and ensure a more logical flow of ideas.**

6. Line 742-750: It is more likely that the $^{13}\delta C$ shift of TOC reflected the source of organic matter rather than the decomposition of OM. Can you get some clue for the quality of organic matter based on the sediment age measured with tracers?

**Response: We appreciate this insightful comment. Distinguishing whether the observed δ¹³C shift in TOC is primarily driven by changes in organic matter sources or by selective decomposition remains challenging. At PS, within the zone of high hydrogenotrophic activity, the low and surface-like C/N ratios suggest the persistence of relatively labile OM, rather than a major change in the OM source. However, lipid biomarker profiles, particularly the concentration of long-chain fatty acids, indicate a shift toward more autochthonous OM. We have clarified this in the revised manuscript as follows: "The δ¹³C shift of TOC within the most active zone at PS likely reflects either the selective loss of isotopically light OM fractions, or changes in organic matter sources, as indicated by depth-related variations in both short- and long-chain fatty acid concentrations." (Lines 746-749)**

7. Line 815-816: This is somewhat not the conclusion supported by the study, I would suggest to revise here.

   **Response: We have revised this sentence to better reflect our findings and to avoid overgeneralization ("Instead, other environmental factors such as substrate availability or microbial community dynamics, may influence methane production, although the specific mechanisms remain uncertain." Line 810-812).**